# Exploration of MPSO-Two-Stage Classification Optimization Model for Scene Images with Low Quality and Complex Semantics

**DOI:** 10.3390/s24123983

**Published:** 2024-06-19

**Authors:** Kexin Liu, Rong Wang, Xiaoou Song, Xiaobing Deng, Qingchao Zhu

**Affiliations:** Department of Information Engineering, Engineering University of PAP, Xi’an 710086, China; wangrong981108@163.com (R.W.); e_miracle@163.com (X.S.); dxb7595@163.com (X.D.); tgzy0516zqc@126.com (Q.Z.)

**Keywords:** low-quality scene image classification, PSO, two-stage classification, semantically ambiguous scenes, outdoor scene classification

## Abstract

Currently, complex scene classification strategies are limited to high-definition image scene sets, and low-quality scene sets are overlooked. Although a few studies have focused on artificially noisy images or specific image sets, none have involved actual low-resolution scene images. Therefore, designing classification models around practicality is of paramount importance. To solve the above problems, this paper proposes a two-stage classification optimization algorithm model based on MPSO, thus achieving high-precision classification of low-quality scene images. Firstly, to verify the rationality of the proposed model, three groups of internationally recognized scene datasets were used to conduct comparative experiments with the proposed model and 21 existing methods. It was found that the proposed model performs better, especially in the 15-scene dataset, with 1.54% higher accuracy than the best existing method ResNet-ELM. Secondly, to prove the necessity of the pre-reconstruction stage of the proposed model, the same classification architecture was used to conduct comparative experiments between the proposed reconstruction method and six existing preprocessing methods on the seven self-built low-quality news scene frames. The results show that the proposed model has a higher improvement rate for outdoor scenes. Finally, to test the application potential of the proposed model in outdoor environments, an adaptive test experiment was conducted on the two self-built scene sets affected by lighting and weather. The results indicate that the proposed model is suitable for weather-affected scene classification, with an average accuracy improvement of 1.42%.

## 1. Introduction

Multimedia resources such as images and videos have become the main media through which people acquire information nowadays [1]. How to quickly find the desired semantics from a large number of redundant scene images has become a popular task in computer vision in recent years. Image scene semantic classification has made contributions to improving work efficiency in various industries, such as rescue agencies which need to determine the type of disaster scene to achieve timely search and rescue in affected areas; public security organs which need to assess the topography of riot scenes to achieve effective control; and transportation agencies which need to judge the distribution of facilities in a short period of time after the planning of paths and so on. However, it is often affected by harsh environmental conditions and the large number of scene elements, along with strong interference, which results in low-quality first-line image information, which is unfavorable for subsequent classification. For example, misclassifying lanes, bridges, embankments, and rural roads can create obstacles for commanders’ decision-making.

Scholars have paid more attention to the setting of the network architecture in order to improve the accuracy of complex semantic scene image classification, constantly innovating methods and achieving certain results [2,3,4]. However, the test objects of the above methods are mostly international standard clear image scene sets, seldom considering the negative impact of low-quality scene images on classification [5]. Even though scholars have gradually discovered that the underlying cause of the decline in classification performance is low-quality images [6,7], the conclusions are mostly applicable to remote sensing images [8] or clear image sets with artificial noise [9], without considering low-quality real-measured images that would be affected by various external factors. For example, network propagation delays can degrade the high-definition quality of images [10], severe weather and dense smoke environments can cause semantic blurring of images [11], poor lighting conditions can lead to misreading of image information [12], etc. These all indicate that it is necessary to explore the causes of low-quality real-measured images and find matching classification methods.

However, scholars have found that even with high-precision CNN architectures, classification performance can still drop significantly when faced with low-quality images [9]. This is because the internal pixel structure of the image neighborhood in the actual environment degrades [13], causing a sharp drop in image resolution and leading to poor image quality [14]. Therefore, it is imperative to add image preprocessing operations [7] to improve the classification effect of low-quality scene image classification models.

To develop a practical model for low-quality scene image classification, this paper makes two innovative contributions: 1. We propose a new method. By refining 21 existing scene classification methods based on two focal points of structural complexity and semantic clarity, a two-stage classification optimization model based on the Mix-optimization PSO Algorithm (MPSO) is proposed. To this end, we introduce an improved hybrid PSO algorithm to optimize the parameters of the Level-1 classification module DCNN-ELM network and the Level-2 reconstruction module SRCNN network in a nested iterative manner, thereby improving the semantic classification effect of low-quality complex scene images. 2. We build a real-measured low-quality dataset. By constructing multi-complex semantic news scene datasets and low-quality datasets under the influence of visible light and environment, the research gap in low-quality image datasets is filled, providing data support for the three different types of experiments with existing scene datasets.

The structure of the paper is arranged as follows: Section 2 summarizes the existing classification methods based on the classification objects. Section 3 introduces the model architecture and principles proposed based on Section 2. Section 4 conducts feasibility and applicability tests. Section 5 focuses on discussing the experimental results and the contributions made by the model, reflecting on the shortcomings and providing prospects. Section 6 concludes the entire paper.

## 2. Related Research Progress

After collecting a large amount of literature, this article summarizes the current scene classification methods into two categories based on structural complexity and semantic clarity.

### 2.1. Development Status of Scene Image Classification Methods Based on Structural Complexity

Due to the rich semantic content of scene images, the structural complexity has a significant impact on classification performance. To illustrate this issue, we leverage the results of the previously published research [3] and use the DCNN-ELM algorithm to perform classification tests on our self-built military weapon image datasets and the low-quality scene datasets built in this paper. The classification results for representative scene images are recorded in Figure 1. It is evident that for images with clear semantic content such as “warplane” and “blast”, the main subject is clearly distinguishable from the background, resulting in a high classification accuracy rate of up to 99%. However, for images with high natural environmental influence such as “forest” and “fire”, which are more susceptible to natural environmental influence, the classification performance drops to 84% due to the difficulty in distinguishing the subject from the background. Below, we will detail scene classification methods for different semantic complexities.

For efficient recognition of rough shapes and geometric structures in single-target scenarios [2], a large number of methods have emerged that extract features first and then combine them with classifiers to achieve recognition. Currently, scholars are constantly innovating in feature extraction methods, such as extracting Gist descriptor underlying features [15], fusing local features [16], or fusing features at different levels [17]. In order to further improve accuracy, the Bag-of-Visual-Words (BoVW) and Gaussian Mixture Model (GMM) have been quickly discovered and applied in scene image classification for their high robustness. In 2024, scholars used the Boruta feature selection algorithm for geomorphic features to distinguish archaeological objects [18]. However, the aforementioned techniques require manual feature extraction and are not widely applicable. They also have semantic ambiguity issues for scenes with cluttered backgrounds. For example, items involved in indoor blind training scenarios are often complex and confusing [19], and outdoor traffic scenes often contain multiple elements such as pedestrians, vehicles, and buildings [20,21]. The similarity and diversity between these image pixels [22,23] make image interference information more abundant, and traditional classification methods lose feature details, resulting in low recognition rates.

For the accurate recognition of complex semantic scenes mentioned above, scholars have frequently innovated around network optimization and parameter adjustment methods, mainly reflected in the methods outlined below.

One approach is to deepen the number of network layers. In order to overcome the problem of insufficient training data, scholars have increased the number of network layers on the basis of convolutional neural network (CNN) architecture to extract more semantic features [22], and have successively developed GoogLeNet [23], the ResNet model [24], and the ResNet model combined with Inception architecture [25] to optimize classification performance. However, multi-layer convolution inevitably loses key information, resulting in limited accuracy.

A second approach is to optimize the network architecture. In 2020, scholars streamlined the classification structure to ensure the classification accuracy and execution speed of indoor scenes [17]. For example, in 2021, scholars combined low-level manual features with deep CNN multi-stage features (HF-MSFs) based on the spatial pyramid structure [19], and improved the architecture to achieve multi-feature adaptive fusion [26] in 2023. In the same year, FcA et al. first applied the upsampling structure to remote sensing image scene classification [8] and achieved good results.

The third approach is to innovate classifiers. In recent years, CNN-ELM based on CNN architecture [11] has been widely applied in image classification with the advantages of high efficiency and high accuracy [27], which also provides ideas for scene image classification in this article.

The fourth approach is to improve the initialization method of network parameters. Scholars have designed a network parameter assignment method [4,6,28] to enhance the accuracy of the classification network. Additionally, in 2023, scholars encoded image segments into sequences and combined them with convolutional architectures to form the Segment Anything Model (SAM) [29] for automatic semantic segmentation of scene images. In 2024, this architecture achieved good results in medical image detection, but its precision for classifying unclear cancer cell images was not high [30].

However, most of the above methods are based on noise-free and clear images, and the high-accuracy classification of low-quality images is still in the initial stage of research.

### 2.2. Progress in Low-Quality Image Classification Methods Based on Semantic Clarity

Low image quality often affects subsequent classification [20,21]. To illustrate this issue, we applied the DCNN-ELM algorithm to classify and compare the same scene using the IC Image Sets and WD Image Sets established in this paper. For example, as shown in Figure 2, the PSNR value of the left image is 0.89 dB higher than that of the right image, clearly indicating that the right image’s edge is deformed due to the inherent resolution of the equipment. In the end, the correct classification rate of the left image is 1.2% higher than that of the right image.

Of course, image quality is not only affected by the inherent resolution of photography equipment, but also by factors such as weather conditions, geographical environment, and time period, which can lead to a decrease in semantic clarity and thus affect classification results. For example, underwater mapping tasks that are subject to the Refraction Effect [30] usually have to take into account multiple factors affecting the image quality such as the position of the capturing camera (above or below water), water refraction, water depth, and light propagation, which undoubtedly have a detrimental effect on the subsequent scene categorization. At present, there are two main optimization schemes for improving the accuracy of low-quality image classification: classification image preprocessing and modifying the classification network structure.

The first type focuses on adding image preprocessing operations before the classification network [31] to enhance image features. Taking semantic fuzzy image classification affected by noise as an example, early scholars made full use of underlying features such as color and texture or HOG features to design filter structures [32] to enhance detail features. However, it is unavoidable to find that an “enhanced image” is not a “clear image”, as there is a gap in feature distribution between the two which cannot benefit from low-quality image classification under the influence of complex scenes [7], nor can it guarantee that similar structural areas are uniformly repaired [13]. Subsequently, the domain adaptation method [33] became mainstream, and the use of neural network architecture became the focus of research.

Scholars have found that image preprocessing methods based on super-resolution reconstruction technology [34] are particularly effective for classification improvement. Taking infrared image classification, which has gradually become a research hotspot since 2020, as an example, feature-based reconstruction methods emerged in 2021 [35], and models with covariant function regularization were proposed in 2022 [36]. However, the above processing operations mostly ignore the optimization of subsequent classification networks, and the design of quantitative indicators is not comprehensive enough.

The second type focuses on optimizing the structure of low-quality image classification networks to avoid the loss of critical information. Taking the limited dataset and difficulty in obtaining high-quality remote sensing image classification as an example, in order to achieve high accuracy classification based on complex spectral features, network deployment was innovated based on CNN architecture in 2018 [37]. Extreme structure DenseNet was launched in 2019 [38], while scholars first used upsampling structure for image scene classification in 2021, and designed DRSNet architecture [8]. Finally, lightweight newCNN was proposed in 2022 [39].

Taking inspiration from the above, on the one hand, image reconstruction pre-operation is introduced in order to better complete the task of self-collected low-quality image classification, and on the other hand, the existing classification network architecture is improved to comprehensively enhance classification accuracy.

## 3. Two-Stage Classification Optimization Algorithm Model Based on MPSO Method

To explore the best approach for classifying low-quality scenes that satisfy complex structures and semantic ambiguity, this paper proposes to construct a two-stage hyperparameter particle swarm optimization strategy based on a convolutional neural network architecture, as shown in Figure 3.

An improved fusion-based particle swarm optimization (MPSO) algorithm is introduced to iteratively optimize the parameters of the Level-1 classification module DCNN-ELM network and the Level-2 reconstruction module SRCNN network. Each stage’s selected particle combination represents one configuration of the optimized integrated network.

Two types of parameters, the size and number of connection weights between the input, output, and implicit layers of the first stage reconstruction network constitute the initial population size. The population in each type of parameter is identified using [*p*_1, *p*_2, *p*_3 …*p*_*m*]. The five elements of the second-stage classification network, namely the number of convolutional layers (consistent with the number of pooling layers), the step size, the convolution sum size, the number of convolutional kernels, and the learning rate, which together constitute the initial population size, are regarded as hyperparameter populations and are identified using [*p*_11, *p*_12, *p*_13 …*p*_1*k*] [*p*_21, *p*_22, *p*_23 …*p*_2*k*] …[*p*_*m*1, p_*m*2, p_*m*3 …*p*_*mk*] for identification. The first stage is based on SRCNN architecture, using Mix-optimization PSO Algorithm (MPSO) group exploration thinking to calculate parameter particle fitness and find the best set of hyperparameters suitable for the current reconstruction network. Then, entering the second stage, based on DCNN-ELM architecture, randomly initialize the network parameters and iterate. When the accuracy is close to 70%, calculate the occurrence probability of each category after docking with ELM. If the current optimal particle fitness value is met, call the ELM layer, calculate the weight of the hidden layer, and cache the β matrix.

In summary, the first stage of the group involves searching space to find the best set of reconstruction hyperparameters, and the second stage of the group uses the sub-search space to find the best set of classification architecture parameters for these reconstructed layers. In this way, the combination of hyperparameters selected after completing each layer cycle constitutes a new classification model. Each section is described in detail below.

### 3.1. SRCNN Structure for Level-1 Reconstruction Stage

Considering the essential influence of image resolution, in order to realize the preprocessing operation of low-quality images, the SRCNN model in Figure 4 is selected for the level-1 reconstruction of this model, and the calculation method is shown in Table 1.

Considering the running time, the initial value is chosen to be 64, the second is chosen to be 32, and the initial values of the three-layer filters $f_{1}\times f_{1}$, $f_{2}\times f_{2}$, $f_{3}\times f_{3}$ are 9×9, 1×1, 5×5, respectively. The learning rate of the first two layers of the network is $\eta_{1}=\eta_{2}{=10}^{-4}$, and the learning rate of the last layer is $\eta_{3}{=10}^{-5}$. The picture channel is chosen to be 1.

### 3.2. DCNN-ELM Structure for Level-2 Classification Stage

After level-1 completes a small cycle of reconstruction network parameter selection for each execution, the DCNN-ELM network architecture is selected and enters level-2 for iterative parameter selection. 

After a small cycle of selecting and reconstructing network parameters is completed in level-1, the parameter iteration based on DCNN-ELM network architecture is carried out in level-2. With the increase in the number of convolution layers, the image pixel value represents more details of the image after being acted upon by the convolution kernel, which is helpful for the judgment of the final result. At the same time, weight sharing in CNN makes the number of parameters only related to the size of the convolution kernel and the number of feature maps. This is reflected in the smaller convolution kernel, which helps to reduce parameter complexity. Therefore, information such as the number of feature maps, the size of the convolution kernel, and the step size constitute the initial population scale in the second stage, and the initialization model setting is set as shown in Figure 5. The calculation method of feature extraction is listed in Table 2.

This model takes the output (n × 1 dimensions) after three times pooling as the input of ELM, and the hidden layer H contains nu neurons, i.e., any node $j\in \left[1,nu\right]$, which corresponds to m neurons in the output layer. Then, for N training samples $i\in \left[1,N\right]$, the mapping relationship between the input feature $X_{input}={\left(x_{i1},x_{i2},\ldots ,x_{\mathrm{in}}\right)}^{T}$ and output label $Y_{output}={\left(y_{i1},y_{i2},\ldots ,y_{\mathrm{im}}\right)}^{T}$ can be calculated by Equation (1):(1)$$Y_{output}=\sum_{j=1}^{nu}\beta_{j}g(W_{j}X_{input}+b_{j})$$
$W_{j}=\left(w_{j1},w_{j2},\ldots ,w_{jn}\right)$ denotes the connection weight of the *n*-th neuron of the input to the *nu*-th neuron of the hidden layer. $\beta_{j}=\left(\beta_{1j},\beta_{2j},\ldots ,\beta_{mj}\right)$ denotes the connection weight of the nu-th neuron of the hidden layer to the *m*-th neuron of the output.

Normally, the actual output is *T*. If the number of neurons in the hidden layer is equal to the number of samples in the training set, the actual output is simplified according to Equation (1) as:(2)$$H\beta =T^{\prime }$$
$T^{\prime }$ is the matrix transpose of T and is the implicit layer output matrix, in the form of Equation (3):(3)$$H={\left[\begin{array}{cccc} g(w_{1}x_{1}+b_{1}) & g(w_{2}x_{1}+b_{2}) & \cdots & g(w_{nu}x_{1}+b_{nu})\\ g(w_{1}x_{2}+b_{1}) & g(w_{2}x_{2}+b_{2}) & \cdots & g(w_{nu}x_{2}+b_{nu})\\ \vdots & \vdots & \vdots & \vdots \\ g(w_{1}x_{N}+b_{1}) & g(w_{2}x_{N}+b_{2}) & \cdots & g(w_{nu}x_{N}+b_{nu}) \end{array}\right]}_{N\times nu}$$
w and *b* are usually kept constant during training, so usually β is the only parameter to be learned in ELM. This requires the introduction of the Moore–Penrose generalized inverse $H^{†}$ to solve for β and obtain $\beta =H^{†}T$, which gives the output after training is complete:(4)$$Y_{output}=g(X_{input})\beta =h(X_{input})H^{†}T$$

The ELM layer is invoked based on the classification accuracy, and the MPSO algorithm is applied in combination with the cache weights to iteratively search for optimization of the population in this loop, in order to complete the optimal parameter combinations of the classification network under the current optimal reconstruction network architecture.

### 3.3. Mix-Optimization PSO Algorithm

As seen in Section 3.1 and Section 3.2, the basic CNN architecture has narrow hyperparameters such as weights, filter sizes, numbers, step sizes, and activation functions, as well as generalized hyperparameters such as the learning rate and the order of the number of layers, which collectively determine the convergence performance of the model and greatly affect the classification accuracy. Given that CNN hyperparameter fine-tuning is a multimodal function optimization problem, scholars currently prefer to combine the Gradient Descent method and optimize the network hyperparameters with the PSO method [40] to improve the classification performance, which is reflected in Equation (5).
(5)$$\left\{\begin{array}{c} v_{m}(t+1)=\omega v_{m}(t)+c_{1}a_{1}\left(pbest_{m}-x_{m}(t)\right)+c_{2}a_{2}\left(gbest_{m}-x_{m}(t)\right)\\ x_{m}(t+1)=x_{m}(t)+v_{m}(t+1) \end{array}\right.$$
where t is the number of iterations, ω is the inertia weight, $r_{1}$ and $r_{2}$ are constants between [0, 1], and *a*1 and *a*2 are the learning factors. $pbest_{m}$ stands for the individual optimum, $gbest_{m}$ stands for the group optimum, $v_{m}\left(t+1\right)$ is the post-update velocity, and $x_{m}\left(t+1\right)$ is the post-update position.

However, it cannot be ignored that the multi-layered CNN network parameters require more particles than the classic particle swarm optimization algorithm, and the complex iteration number leads to a decline in optimization ability, thereby adversely affecting the classification performance. To avoid this problem as much as possible, many kinds of literature have proposed methods to improve the inertia weight (*ω*), social coefficient (*a*1), and cognitive coefficient (*a*2), and have successfully been applied to DCNN [27] and CNN-ELM [5] network classification. Inspired by the existing research [40], we proposed an MPSO optimization mode based on the iteration combination, which on the one hand weakens the group cognition in the early stage of the iteration cycle and strengthens the overall search ability later, accelerates the adjustment of the learning factor in real-time and correcting the particle optimization trajectory in a timely manner. On the other hand, we optimized the inertia weight in a stage-wise manner to assist the learning rate. The strategy is as follows:

① Changing learning factors

(6)$$\left\{\begin{array}{c} a_{1}=a_{1\max}+\left(a_{1\max}-a_{1\min}\right)*\left(\frac{t}{t_{\max}}\right)\\ a_{2}=a_{2\max}+\left(a_{2\max}-a_{2\min}\right)*\left(\frac{t}{t_{\max}}\right) \end{array}\right.$$
where $a_{1\max},a_{1\min}$ are often the maximum and minimum values of $a_{1}$, which are taken as 2.5 and 0.5, respectively. $a_{2\max},a_{2\min}$ are the maximum and minimum values of $a_{2}$, which are taken as 2.5 and 0.5, respectively. t is the current number of iterations, and $t_{\max}$ is the maximum number of iterations.

② Optimizing inertia weights


(7)
$$\omega \left(t\right)=\left\{\begin{array}{cc} 0.9 & t<0.2t_{\max}\\ \frac{1}{1+e^{\left(10t-2t_{\max}\right)/t_{\max}}} & otherwise \end{array}\right.$$


The initial population size at the time of the experiment was 40, and the spatial dimensions were consistent with the weight threshold.

## 4. Experiment Results

As shown in Figure 6, the experiment mainly consists of two modules. The first module mainly tests the feasibility of the proposed architecture. For the existing scene set and the scene set formed by intercepting low-quality news keyframes, the rationality of the classification structure and the necessity of the pre-reconstruction link are verified, respectively, so as to comprehensively evaluate the classification adaptability and high accuracy of the model in this paper.

The second module mainly evaluates the applicability of the model. For the low-quality image set of news scenes and the self-built low-quality image set affected by the environment, the application potential of the model is explored by realizing multi-scene semantic classification with different clarity.

### 4.1. Exploring the Rationality of Classification Model Architecture

In this part, three types of datasets are selected as shown in Table 3, and the validation set and test set are set according to 2:8.

Moreover, we have completed the fitness evaluation of our model on the 15-scene dataset and selected an appropriate batch size value as in Figure 7 to demonstrate the advantages of introducing MPSO for parameter adjustment.

Generation refers to the number of genetic cycles, and accuracy rate represents the average accuracy rate after classification. We found that a high accuracy rate can be maintained under the condition of fewer parameters, and the accuracy rate tends to stabilize after seven generations of cycles. However, the accuracy rate starts to decrease with the growth in batch size, and the accuracy rate tends to be optimal in the genetic cycle when the batch size is chosen to be 25.

#### 4.1.1. Intuitive Comparison of Multi-Method Classification Effects

In order to more comprehensively illustrate the applicability and classification performance of the classification structure proposed in this paper, we roughly classify the 20 scene classification methods mentioned in the previous section into three major categories: feature extraction improvement, network architecture updating, and classifier design. We carry out comparative experiments with the methods in this paper, and record the classification results in Table 4.

For the 15-scene dataset, the algorithm proposed in this paper is 1.54% higher than the best existing method, ResNet-ELM. For the MIT-Inoor dataset, the algorithm proposed in this paper is 0.18% higher than the best available method Places365-VGG. For Sun datasets, although 0.1% lower than the best available method CNNang-SVM, the comprehensive advantage remains ahead. After an in-depth analysis of the reasons, we found the following results:

Accuracy improves with a decrease in the number of classification labels. Although a large dataset helps to improve the classification, the accuracy decreases when the variety of classification labels increases. Therefore, the 15-scene dataset with fewer labels has more advantageous classification accuracy compared to the other two datasets.

Along with the increase in structure complexity, the accuracy decreases. The overall effect is that the recognition accuracy of indoor scenes is lower than that of outdoor scenes. We attribute this to the fact that indoor scenes contain too many trivial structures, which are semantically rich and complex and adversely affect feature judgments. Optimizing the classifier is more conducive to improving the classification effect of the outdoor scene dataset; improving the classification network architecture is more advantageous in classifying the indoor scene dataset. Our method still maintains excellent performance for complex scenes due to the advantages of combining the above two approaches: initializing the network weights using swarm intelligence algorithms, and streamlining the network structure against ELM to extract more detailed features.

#### 4.1.2. Comparison of Classification Effect of Multi-Semantic Labels

We take 15 scenes with excellent classification performance as an example. The number of each tag is recorded in Figure 8, and the classification results are recorded in the form of a confusion matrix in Figure 9. The comprehensive analysis reveals that:

Consistent with the previous conclusions, the proposed architecture achieves good classification results in outdoor scenes such as “coast”, “forest”, and “suburb”, while it is inferior in indoor scenes with high confusion levels such as “kitchen”, “living room”, and “bedroom”.

Meanwhile, we were surprised to find that the “suburb” scene dataset, although smaller in number than the other scene sets, generally has a high image resolution, which contributes to a classification accuracy of 99.99%. This also means that datasets with higher classification accuracy tend to have higher PSNR values. Inspired by this, we propose to think about whether image resolution affects classification performance, i.e., does the addition of a pre-reconstruction session contribute to higher classification accuracy?

### 4.2. Experiments on the Necessity of Adding a Pre-Reconstruction Architecture

#### 4.2.1. The Effect of Low-Quality Images (Low Resolution) on the Degree of Classification Performance

Due to the disadvantage of the limited number of existing low-quality datasets, we reduce the resolution of the 15-scene dataset images, i.e., select different upscale factors to realize the Bicubic interpolation operation. Figure 10 shows the visual effects and PSNR values of outdoor scenes “coast”, “highway”, “open country”, “forest”, and indoor scenes “living room” and “bedroom” under different upscales.

The PSNR values of outdoor scenes with high classification accuracy are generally higher than those of indoor scenes. Based on semantic complexity, the 15 scenes are roughly divided into three categories: indoor, outdoor, and mixed indoor-outdoor scenes. The proposed method is used to classify them, and the results are recorded in Table 5.

As the value of upscale increases, the classification performance continuously declines. When upscale is set to 3, the accuracy decreases by 0.37% compared to when upscale is set to 2, and when upscale is set to 4, the accuracy decreases by 0.53% compared to when upscale is set to 3. This shows that as the image resolution decreases, the loss of key information gradually increases the impact on classification accuracy.

Upon further investigation, it was found that the degradation of outdoor image classification performance after degradation was higher than that of indoor images. For example, when the upscale was changed from 2 to 3, the outdoor image classification performance decreased by 0.43% while the indoor image classification performance decreased by 0.25%. To investigate the impact of low-resolution scene images on classification performance, we recorded the classification results at different resolutions in Figure 11.

From the figure, it can be concluded that the impact of degradation on semantic complexity and confusing scenes is greater than that on scenes with a single structure. In particular, the impact on outdoor scenes that are easily affected by the environment (such as “coast” and “open country”) is greater than that on indoor scenes (such as “office” and “living room”). We hypothesize that this is due to the omission of typical features and loss of edge details in the extraction process of low-resolution images. For example, “coast” and “open country” are easily confused, which shows the importance of adding pre-reconstruction operation before classification.

#### 4.2.2. The Degree of Contribution of the Pre-Reconstruction Structure to the Classification Performance Improvement in This Paper

This part of the research object comes from 1068 short videos on the CCTV website, from which 4861 keyframes are selected and categorized into “Stage”, “Conference”, “Woodland”, “Fire”, “Building”, “Flood” and “Highway”. The specific composition and quality ratio are given in Table 6. The test set and training set are divided according to the ratio of 2:8, and the input size is uniformly 256 × 256.

After the performing degradation process (upscale taken as 2, 3, and 4) for the above image set, based on the classification architecture proposed in this paper, six different preprocessing methods were compared, including Bilateral Filter, dehazing method, SC, ANR, SRCNN, and MPSO SRCNN. The classification results are recorded in Figure 12.

From Figure 12, it is intuitive to find that the introduction of image preprocessing operations improves the classification accuracy of low-quality images. The contribution rate of the reconstruction-based preprocessing method to the improvement of classification network accuracy is generally higher than that of the image enhancement method, especially the method proposed in this paper, which has the best effect. At the same time, it was found that when upscale is set to 3, the classification accuracy of the proposed method can be significantly different from that of other methods, so upscale should be set to 3 in subsequent research.

In addition, the classification improvement rate of the model in this article for outdoor scenes is higher than that for indoor scenes. We speculate that this is because outdoor scenes are often affected by weather conditions and lighting conditions, resulting in poor image quality. The pre-reconstruction step of the model in this article helps to improve classification performance. Inspired by this, we ask whether the model can achieve improvement in classification performance for low-quality images taken in real-life scenarios.

### 4.3. Application Test of the Model for Low-Quality Complex Scene Classification

#### 4.3.1. Exploration of the Types of Low-Quality Images Suitable for the Proposed Model

We conducted experiments for seven news scenes divided into two categories of better and worse according to their semantic clarity, using or not using pre-reconstruction as a comparison, and recorded the results in Table 7.

From Table 7, we can see that the semantic classification accuracy of the better scene was improved by 1.53%, and the semantic classification accuracy of the poor scene was improved by 2.11%. It is obvious that the improvement in low-quality images is greater, especially for the “fire” and “building” semantic categories, which were improved by 3.82% and 4.53%, respectively. The advantage of semantic classification for the “stage” semantic category in indoor images is higher than that for the “Conference” semantic category with good lighting. This shows that introducing preprocessing operations is more effective for image classification in scenarios affected by lighting and environment. Therefore, in the next part of the experiment, we will focus on studying the value of the proposed model in classifying low-quality images collected under weather and lighting conditions.

#### 4.3.2. Exploration of the Application Value of the Proposed Model in Improving Classification Performance

This part of the experiment is conducted for the special environment self-built low-quality image dataset (the composition is shown in Table 8). IC Image Sets represent the infrared image set built by the professional infrared camera after being affected by illumination change, with a total of 1087 pictures. WD Image Sets represent the image sets created by cell phones after being affected by weather disturbances (rain, snow, haze), totaling 1174 images.

Firstly, the visual effect of using different algorithms in the pre-reconstruction stage of the model is recorded in Figure 13 and Figure 14, and it is clearly found that the reconstructed network pattern in this paper can better emphasize the detailed features. In particular, the semantic reconstruction of “Building” has obvious effects. In the local zoomed-in image, it is found that the window outline and the building structure have been repaired, which is more helpful for the subsequent categorization. The “Woodland” semantic shows better performance in IC Image Sets, and the edges of leaves are sharpened to a higher degree. The “Traffic” semantics show superior performance in WD Image Sets, with wheels and street lights being reconstructed to be more readable.

Secondly, in Figure 15, the experimental results using the two-stage classification model proposed in this paper versus the classification-only model are documented. The specific improvement is recorded in Table 7.

From Figure 15, it is found that the model introducing the reconstruction phase improves the correct rate of semantic classification for both datasets. The improvement of semantic classification accuracy for poor-quality scenarios is stronger than that for better-quality scenarios, especially for the discrimination of “Building”, which is consistent with the findings in Figure 13 and Figure 14, and shows that the reconstruction stage contributes greatly to the improvement of classification performance.

From Table 9, it is found that for better quality images, IC-Image Sets have better classification results with an average improvement of 0.67%. For poor-quality images, WD-Image Sets have better classification performance with an average improvement of 1.42%. In general, the semantic improvement of poor-quality scenes is higher: for the “Building” scene, the classification accuracy of IC-Image Sets and WD-Image Sets is improved by 2.08% and 3.52%, respectively. For the “Traffic” scenario, WD-Image Sets are more suitable, with an improvement of 0.67%. For the “Woodland” scenario, IC-Image Sets are more effective, with an improvement of 0.92%.

## 5. Discussion

The comprehensive experiments conducted in this study demonstrate that the proposed two-stage classification model effectively enhances the classification performance for the measured low-quality scene images.

Firstly, we focused on the feasibility of the classification stage model. Consistent with the focus of Section 2.1, we selected three general scene datasets, including indoor and outdoor scenes, based on the complexity of scene structures. We not only demonstrated the architectural stability through parameter selection but also proved its feasibility through comparisons with existing methods. The experimental results reveal that the proposed architecture excels in datasets with fewer classification labels and a higher proportion of outdoor scenes. Additionally, it exhibits higher accuracy in classifying images with higher resolutions. This observation prompts us to consider the contribution of improving the resolution of low-quality images to enhance the overall classification accuracy of the model.

Secondly, we investigated the rationality of the pre-reconstruction stage model. Aligning with the focus of Section 2.2, we established seven news scene image sets based on the causes of low image quality. By integrating diverse image quality enhancement techniques into the unified classification architecture outlined in this paper, we aim to underscore the superiority of the proposed reconstruction model. The results indicate that the proposed architecture achieves better classification accuracy for low-quality scene images affected by environmental and lighting factors. Moreover, the improvement is more significant for images with lower PSNR values. This finding suggests that we should explore the application potential of the proposed model by testing it on practical low-resolution images.

Finally, we tested the practicality of the two-stage classification model. Using equipment to capture and establish IC and WD Image Sets, we conducted comparative experiments between the two-stage and single-stage methods for low-resolution images, employing fewer labels based on the aforementioned results. Both visually and quantitatively, the experimental data clearly demonstrates the practical value of the proposed model.

Overall, the key advantage of the method presented in this paper lies in its innovative integration with practical low-resolution scene images to conceive an optimal classification model. A series of validation experiments fully demonstrate its application capabilities, complementing the test results of practical low-quality image research datasets. This undoubtedly provides a foundation for the analysis of recorded videos containing a large number of image frames. For instance, in emergency rescue missions, UAVs often capture live videos in low-light or harsh environments, where a classification model is urgently needed to assist commanders in quickly identifying critical lifeforms from complex images.

However, it is also recognized that the proposed method has limitations, primarily due to its heavy reliance on semantic elements in input images. If the effective or target information in an image is too small while the interfering information is excessive, it can lead to misclassification and semantic confusion. To address this issue, future research should focus on introducing practical dynamic video target detection, considering the target information in consecutive keyframes for comprehensive judgment. Additionally, optimizing the two-stage classification effect by combining the advantages of the SAM architecture for scenes with distinct edge delineation and achieving automated segmentation are also key areas for further work.

## 6. Conclusions

This paper focuses on the problem of semantic image classification of low-quality complex scenes, and proposes the MPSO two-stage classification optimization model, which is found to have the following advantages through three sets of experiments: 1. The classification effect of the scenes with a few kinds of complex semantics is better than the existing methods. 2. The optimized network design of the pre-reconstruction module greatly improves the classification effect of low-quality scene images. 3. The optimized network design of the pre-reconstruction module still maintains good application potential even when classifying outdoor scenes with large environmental influences. However, there still exists a scene with a lot of useless interfering information and low utilization of useful information. The next step is to focus on how to remove a large amount of interfering information in the pre-reconstruction module to help realize efficient recognition.

## Figures and Tables

**Figure 1 sensors-24-03983-f001:**
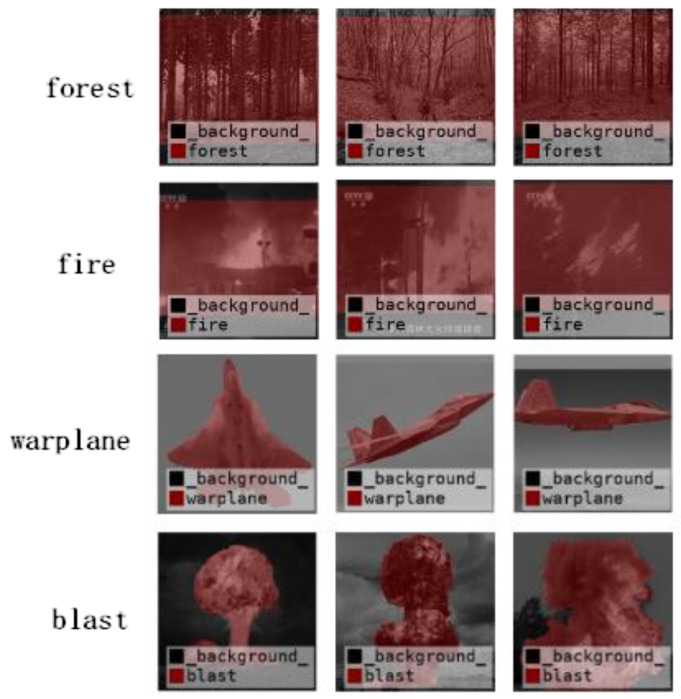
Semantic classification performance for images with different scene complexity.

**Figure 2 sensors-24-03983-f002:**
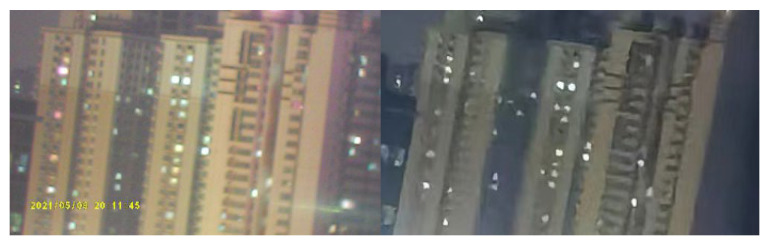
Visual comparison of the same scene taken by different shooting devices.

**Figure 3 sensors-24-03983-f003:**
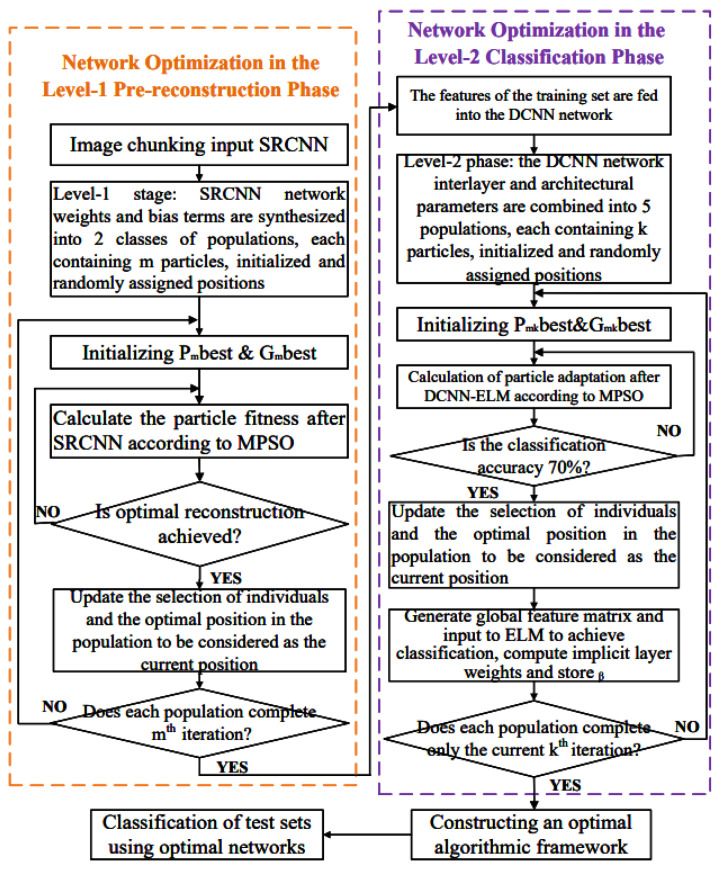
Flowchart of the algorithm.

**Figure 4 sensors-24-03983-f004:**
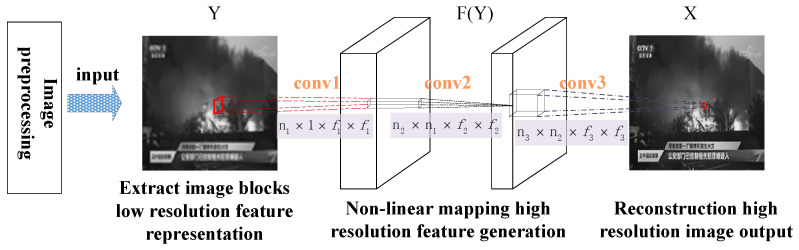
SRCNN architecture.

**Figure 5 sensors-24-03983-f005:**
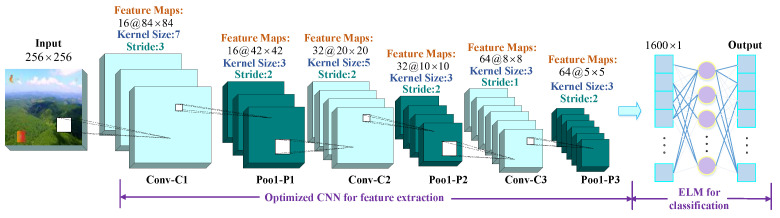
DCNN-ELM initialization model.

**Figure 6 sensors-24-03983-f006:**
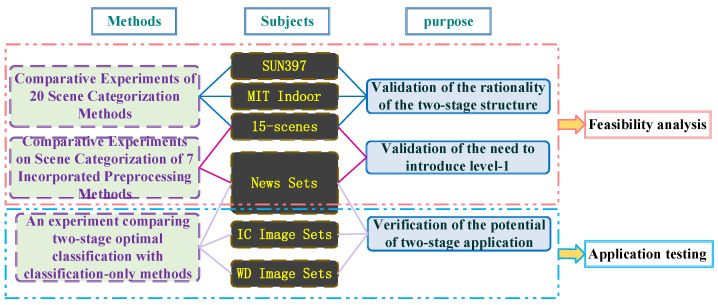
Experimental module design.

**Figure 7 sensors-24-03983-f007:**
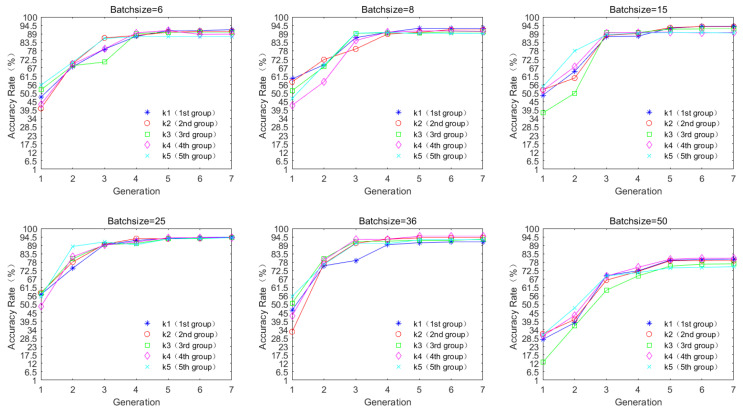
Effect of batch-size selection on network optimal fitness (classification accuracy).

**Figure 8 sensors-24-03983-f008:**
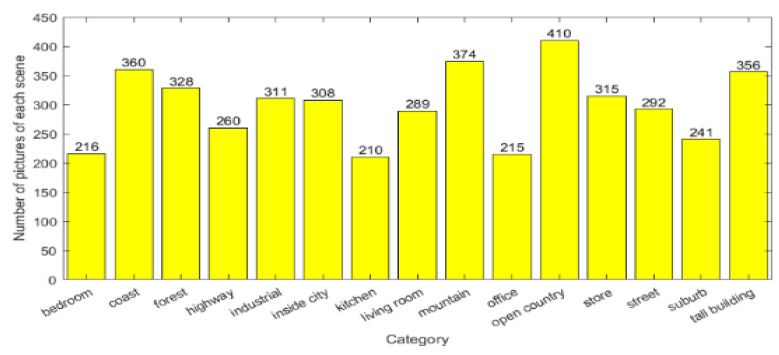
Scene Category and Quantity.

**Figure 9 sensors-24-03983-f009:**
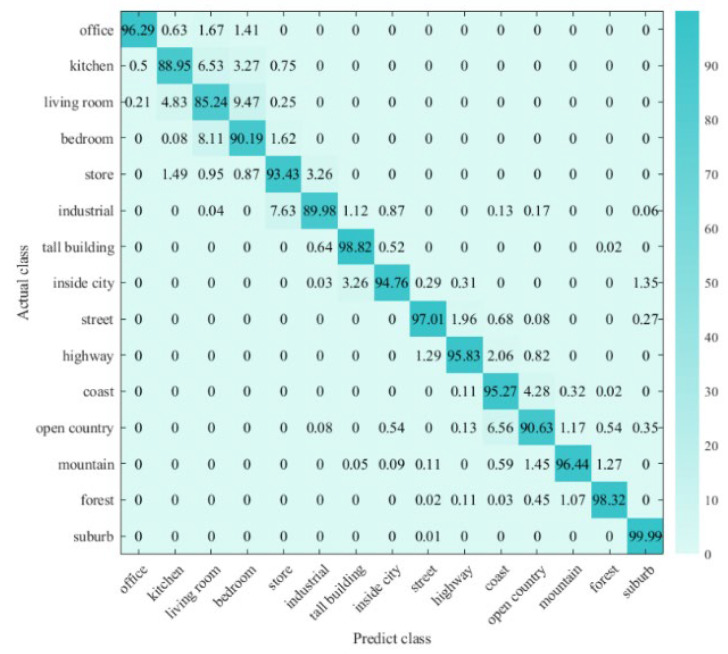
Classification Obfuscation Matrix of 15-scene Dataset.

**Figure 10 sensors-24-03983-f010:**
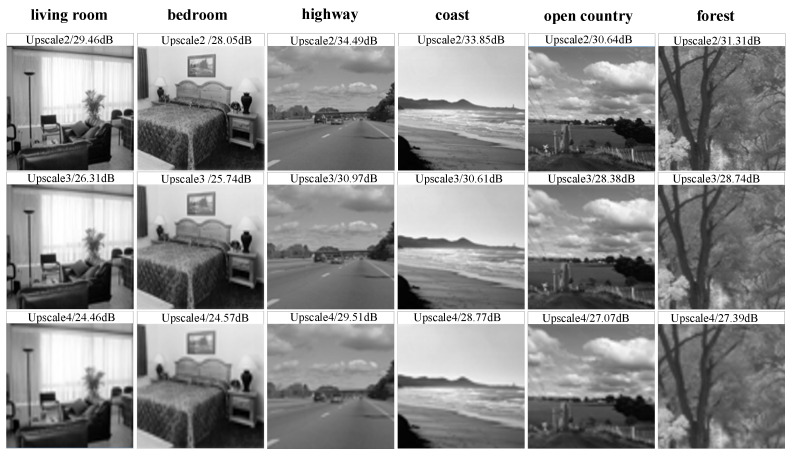
Comparison of image quality affected by inherent limitations of equipment.

**Figure 11 sensors-24-03983-f011:**
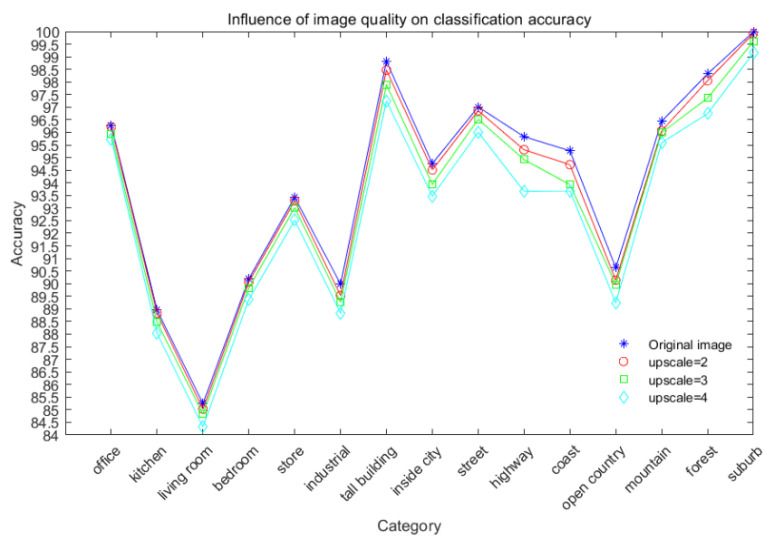
The influence of image semantic quality on classification accuracy.

**Figure 12 sensors-24-03983-f012:**
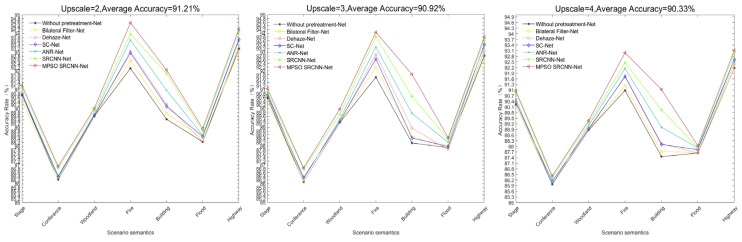
Classification of low-quality news images using different preprocessing methods based on the classification network proposed in this article.

**Figure 13 sensors-24-03983-f013:**
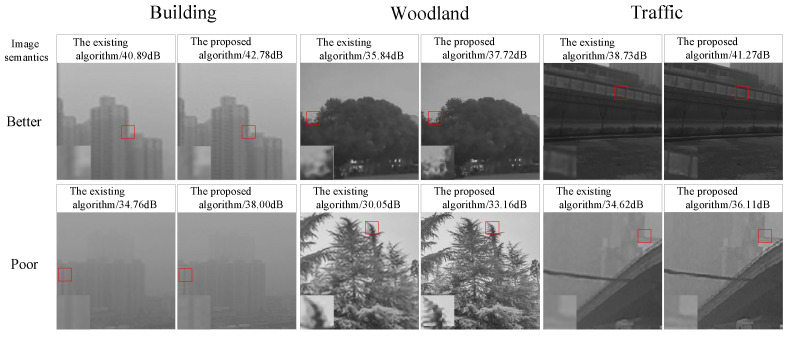
Pre-reconstruction stage visualization of IC Image Sets.

**Figure 14 sensors-24-03983-f014:**
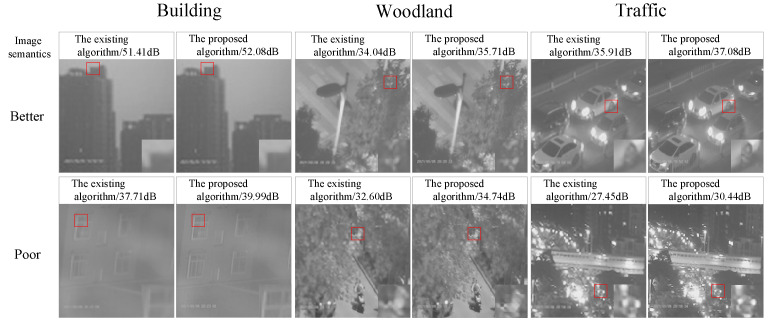
Pre-reconstruction stage visualization of WD Image Sets.

**Figure 15 sensors-24-03983-f015:**
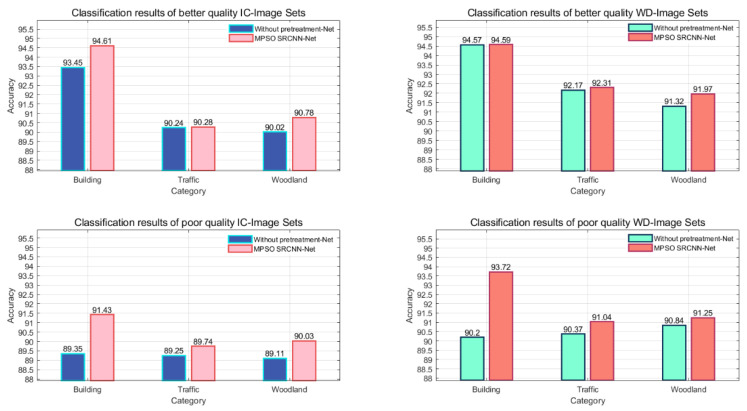
Comparison of classification results with and without the introduction of a two-stage classification model.

**Table 1 sensors-24-03983-t001:** SRCNN calculation flow.

Calculation Step	Calculation Method
1. Image block extraction and feature representation: a set of trained bases (e.g., Haar, DCT, etc.) are used to represent the input extracted image blocks, and a set of optimized filters associated with the bases are used to convolve the image.	$F_{1}(Y_{i})=h(W_{1}*Y_{i}+b_{1})=\max(0,W_{1}*Y_{i}+b_{1})$
2. Nonlinear mapping: the first layer is represented as an n_1_-dimensional vector and is transformed into an n_2_-dimensional vector by nonlinear mapping.	$F_{2}(Y_{i})=\max(0,W_{2}*F_{1}(Y_{i})+b_{2})$
3. Image reconstruction: the high-resolution feature blocks extracted from the convolutional layer are aggregated, and then the filtering of the feature map is realized in the pattern of averaging.	$F(Y_{i})=W_{3}*F_{2}(Y_{i})+b_{3}$

**Table 2 sensors-24-03983-t002:** Calculation steps for feature extraction.

Calculation Step	Calculation Method	Parameter Meaning
1. Convolutional layer calculation	$x_{ij}^{k+1}=R\left[\sum_{K_{l}}\sum_{p=1}^{P_{i}}\sum_{q=1}^{Q_{i}}x_{K_{l}(si+p,sj+q)}^{k}\omega_{K_{l}(p,q)}^{k+1}+b_{ij}\right]$	$K_{l}$ denotes the number of feature maps connected to the kth convolutional layer, i.e., the number of channels. $P_{i}$ and $Q_{i}$ are the currently set filter width and height, and S is the sliding step. $\omega_{K_{l}(p,q)}^{k+1}$ denotes the value of the convolutional kernel connected to the *k* + 1st layer of feature maps located at (p, q). R is chosen to be the nonlinear excitation function ReLUs. $b_{ij}$ is the bias term.
2. Calculation of pooling layer	$X_{ij}^{k+1}=\max\left\{X_{\left(i,j\right)}^{k},X_{\left(i+1,j+1\right)}^{k},\ldots ,X_{\left(i+Pi,j+Q_{i}\right)}^{k}\right\}$	

**Table 3 sensors-24-03983-t003:** Listing of datasets.

Datasets	Total	Category	Number of Each Category	Data Content
15-Scene	4485	15	200~400	4 indoor; 9 outdoor; 1 mixed indoor-outdoor
SUN397	108,754	397	>100	175 indoor; 220 outdoor; 2 mixed indoor-outdoor
MIT Inoor	15,620	67	>100	67 indoor; five categories: “Store”, “Home”, “Public spaces”, “Leisure”, and “Working place”.

**Table 4 sensors-24-03983-t004:** Comparison of scene classification effects of 21 methods.

Dataset		15-Scene	SUN	MIT-Inoor
Optimization method based on feature extraction improvement	SIFT-SPM	81.4	-	-
	SIFT-GAIJIN	-	-	-
	PlaceNet	-	50.36	68.24
	MOP-CNN	-	51.98	
	CENTRIST + SVM	-	-	36.9
	BOP	-	-	46.1
Optimization methods based on network architecture improvement	ImageNet-AlexNet	84.05	42.61	56.79
	ImageNet-GoogleNet	84.95	43.88	59.48
	ImageNet-VGG	86.28	48.29	64.87
	VGG-16	-	56.2	75.67
	Places365-AlexNet	89.25	56.12	70.72
	Places365-GoogleNet	91.25	58.37	73.30
	Places365-VGG	91.97	63.24	76.53
Optimization methods based on classifier replacement	GoogleNet-SVM	92.09	-	-
	GoogleNet-ELM	92.18	-	-
	ResNet-SVM	93.4	-	-
	ResNet-ELM	92.64	-	-
	CNNang-SVM	-	69	58.4
	CNN-SVM	-	55.14	-
	DCNN-ELM	-	-	69.27
Associative convolutional architecture	SAM	-	-	70.26
The methodology proposed in this paper		94.18	68.9	76.71

**Table 5 sensors-24-03983-t005:** Comparison of classification performance for multi-class scene images of different quality (%).

	Original Image	Upscale = 2	Upscale = 3	Data Content
Indoor (4 types)	89.78	89.63	89.38	88.97
outdoor (9 types)	96.10	95.75	95.32	94.74
mixed indoor-outdoor (2 types)	91.72	91.42	91.16	90.69
All 15 types	94.18	93.88	93.51	92.98

**Table 6 sensors-24-03983-t006:** Composition of Low-Quality News Image Scene Set.

	Semantics	Stage	Conference	Woodland	Fire	Building	Flood
Number	725	698	640	720	703	659	716
Better	350	334	320	324	346	315	355
Poor	375	364	320	396	357	344	361
Number	725	698	640	720	703	659	716

**Table 7 sensors-24-03983-t007:** Classification of News Scene Images with Different Semantic Clarity (%).

Semantics		Stage	Conference	Woodland	Fire	Building	Flood	Highway	Avg
better	Without pretreatment-Net	90.70	86.14	90.13	93.22	90.46	88.67	93.50	90.44
	MPSO SRCNN-Net	91.12	86.89	90.84	95.79	94.73	89.25	94.82	91.97
poor	Without pretreatment-Net	90.24	86.03	87.49	88.26	85.59	86.99	92.19	88.15
	MPSO SRCNN-Net	90.92	86.80	90.95	92.08	90.12	88.03	92.74	90.26

**Table 8 sensors-24-03983-t008:** Self-constructed low-quality datasets with different semantic clarity.

Semantics	IC Image Sets			WD Image Sets		
	Sum	Better	Poor	Sum	Better	Poor
Woodland	271	135	136	263	130	133
Traffic	366	175	191	351	162	189
Building	450	200	250	560	250	310

**Table 9 sensors-24-03983-t009:** Specific enhancements (%).

		“Building”	“Traffic”	“Woodland”	Avg
IC-Image Sets	Better quality	1.15%	0.04%	0.76%	0.67%
	Poor quality	2.08%	0.49%	0.92%	1.28%
WD-Image Sets	Better quality	0.02%	0.14%	0.65%	0.35%
	Poor quality	3.52%	0.67%	0.41%	1.42%

## Data Availability

All relevant data are within the paper.

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
