# Peer review of "Exploration of MPSO-Two-Stage Classification Optimization Model for Scene Images with Low Quality and Complex Semantics"

_sensors, 2024, doi:10.3390/s24123983_

Round 1
Reviewer 1 Report
Comments and Suggestions for Authors
Author Response
Thank you very much for taking the time out of your busy schedule to carefully review our paper and for your valuable comments, your feedback is crucial to our research work. To this end, we have reshaped the “Introduction” section, added a “Discussion” section, improved the “Related research progress” and “Experiment” sections, and revised the “Method” section. Please see the attachment for the specific modifications. :)

Reviewer 2 Report
Comments and Suggestions for Authors
This paper presents a novel approach to enhancing image processing and classification techniques. The authors’ unique presentation style is commendable. However, the structure of the article deviates significantly from the conventional format of Introduction - Methods - Results - Discussion - Conclusion. Various sections seem to be intermingled, making it difficult to discern the paper’s key contributions.
I strongly recommend refining the paper’s structure. While the main sections don’t necessarily need to adhere to traditional naming conventions, it’s crucial that readers can easily understand what part of the paper they’re reading at any given time.
Furthermore, the discussion section needs to be developed or enhanced. The authors should consider addressing the following topics: the implications of their research for the broader scientific community, how their work addresses existing gaps in the field, the limitations of their research, and recommendations for future work.
Lastly, there are specific comments included in the attached PDF file. Incorporating these suggested changes will greatly enhance the quality of the manuscript. I believe the manuscript would greatly benefit from these revisions.

Author Response

(The authors gave the same response as above.)

Round 2
Reviewer 2 Report
Comments and Suggestions for Authors
Thank you for your work. I would like to see this paper published.